# Effects of Daily Physical Activity on Exercise Capacity in Chronic Obstructive Pulmonary Disease

**DOI:** 10.3390/medicina60071026

**Published:** 2024-06-21

**Authors:** Marina Aiello, Annalisa Frizzelli, Roberta Pisi, Rocco Accogli, Alessandra Marchese, Francesca Carlacci, Olha Bondarenko, Panagiota Tzani, Alfredo Chetta

**Affiliations:** 1Department of Medicine and Surgery, University of Parma, 43126 Parma, Italy; 2Cardio-Thoracic and Vascular Department, University Hospital of Parma, 43125 Parma, Italy; 3Otorhinolaryngology Department, Kharkiv National Medical University, 61022 Kharkiv, Ukraine

**Keywords:** physical activity, exercise capacity, COPD

## Abstract

*Background and Objectives*: In adults, 150 to 300 min a week of moderate-intensity physical activity is the recommended daily level to maintain or improve fitness. In subjects with chronic obstructive pulmonary disease (COPD), reductions in daily physical activity (DPA) amounts are related to clinically significant outcomes. In this study, we ascertain whether or not COPD patients, when clustered into active (DPA ≥ 30 min a day, 5 days a week) and inactive (DPA < 30 min a day, 5 days a week), may differ in exercise capacity, as assessed by a cardiopulmonary exercise test (CPET). *Materials and Methods*: A large sample of clinically stable COPD patients was retrospectively recruited and then underwent spirometry and an incremental ramp protocol 5–15 watts/min CPET. DPA was assessed by a questionnaire. *Results*: A total of 83 (female 25%, age range 41–85 y) active and 131 (female 31%, age range 49–83 y) inactive participants were enrolled. They were similar in age, sex distribution, body mass index (BMI) and in spirometry. The two groups were significantly different in dyspnea on exertion, as assessed by the modified Medical Research Council (mMRC), and in cardio-metabolic parameters, but not in ventilatory ones, as confirmed by the CPET. *Conclusions*: COPD patients experiencing physical activity of at least 30 min a day, 5 days a week, showed a greater exercise capacity and an improved cardiovascular response to exercise, when compared to inactive ones. Active and inactive participants did not differ in terms of airflow obstruction severity as well as in dynamic hyperinflation and ventilatory inefficiency during exercise. This study further suggests the benefits of regular physical activity in COPD.

## 1. Introduction

In order to maintain or improve fitness, 150 to 300 min per week of moderate-intensity physical activity is the recommended daily level in adult individuals [1]. Furthermore, regular physical activity reduces the risk of developing a large number of chronic diseases and conditions and is valuable in the treatment of numerous diseases [1]. In subjects with chronic obstructive pulmonary disease (COPD), reductions in daily physical activity (DPA) amounts are related to clinically significant outcomes, such as poor health status [2] and reduced survival [3]. A decrease in DPA is not restricted only to the patients with severe COPD but has also been reported in patients with mild COPD [4].

Exercise capacity is the maximum amount of physical exertion that a patient can sustain [5]. It represents the functional capacity of the cardiopulmonary system and is expressed as the maximum rate at which oxygen can be used during maximal exercise, thereby being a function of both cardiopulmonary performance and the maximum capability to remove and use oxygen from the blood [5]. COPD patients commonly show a decrease in exercise capacity mainly because of a ventilatory limitation, even if cardiovascular limitation and/or limb muscle fatigue can occur [6].

In this study, we ascertain whether or not COPD patients, when clustered into an active group (DPA ≥ 30 min a day, 5 days a week) and inactive one (DPA < 30 min a day, 5 days a week), may differ in exercise capacity and in cardiovascular and ventilatory responses to exercise, as assessed by a cardiopulmonary exercise test (CPET). The findings of this study were previously reported in a conference abstract [7].

## 2. Materials and Methods

### 2.1. Patients and Pulmonary Function Testing

In this retrospective study, we consecutively enrolled all patients with COPD, as defined by the criteria of the Global Initiative for Chronic Obstructive Lung Disease [8], who underwent a pulmonary rehabilitation program from January 2014 to December 2023. Inclusion criteria consisted of (1) being in a stable clinical condition (without exacerbation for at least one month); (2) must not have respiratory failure; (3) must not have any comorbidity affecting exercise performance (neuromuscular impairments, anemia, chronic cardiac failure or malignancies); (4) ability to perform a symptom-limited CPET. The protocol was approved by the local ethical committee on 23 March 2021 (final protocol no. 16696, 19 April 2021). All patients signed the written informed consent. The study was conducted in accordance with Good Clinical Practices and the Declaration of Helsinki.

COPD patients were classified into 4 stages (mild, moderate, severe and very severe), based on the predicted value of the forced expiratory volume at 1st second (FEV_1_) (>80%, 50–80%, 30–49% and <30%) [8]. For all patients, we recorded at baseline: anthropometric variables (age, sex and body mass index—BMI, in kg/m^2^) and daily living dyspnea related to physical activity evaluated by the Italian version of the 5-point Medical Research Council scale modified by the American Thoracic Society (mMRC) [9]. DPA was assessed by a questionnaire, which is the Italian version of the “Rapid Assessment of Physical Activity” [10]. Briefly, patients reported whether or not they experienced regular physical activity. Leisurely walking, stretching, light yard working were considered as light physical activities, while brisk walking, cycling, dancing, gently swimming, going up and down stairs as moderate-intensity physical activities as well as jogging, playing tennis as vigorous ones. Duration of daily physical activity (min a day) and weekly frequency of physical activity (day a week) were also recorded. We categorized as active patients those patients who reported at least 150 to 300 min a week of moderate-intensity physical activity.

We performed pulmonary function tests according to international recommendations [11].

A flow-sensing spirometer and a body plethysmograph connected to a computer for data analysis (Vmax 22 and 6200; Sensor Medics, Yorba Linda, CA, USA) were used for these measurements. Total lung capacity (TLC), vital capacity (VC), inspiratory capacity (IC), FEV_1_ were recorded in liters and FEV_1_/VC and IC/TLC ratios were calculated. Lung diffusion capacity for carbon monoxide (DLCO) was measured by the single-breath method using a mixture of carbon monoxide and methane. TLC, VC, FEV_1_ and DLCO were also expressed as a percentage of the predicted values [12].

### 2.2. Cardiopulmonary Exercise Test

CPET was performed through the standardized procedures referring to the incremental exercise test with the supervision of an expert pulmonologist (PT) [13]. Briefly, after calibrating the oxygen and carbon dioxide analyzers and flow mass sensor, participants sat on an electromagnetically braked cycle ergometer (Corival PB, Lobe Bv, Groningen, The Netherlands). After an initial rest of 3 min, patients underwent unloaded cycling for another 3 min with an increment of 5–20 watts for every minute (according to functional status of the patient), to achieve a total time of 8–12 min. The pedaling frequency (patients were instructed to maintain 60 rotations/min, rpm) was reported by a digital display on the monitor of the ergometer.

Breath-by-breath oxygen uptake (V’O_2_ in mL/kg/min), carbon dioxide production (V’CO_2_ in mL/kg/min), tidal volume (VT in L), respiratory rate (RR in bpm) and minute ventilation (VE in L/min) were recorded during the test (CPX/D; Med Graphics, St. Paul, MN, USA). Participants were continuously assessed by a 12-lead electrocardiogram (Welch Allyn CardioPerfect, Delft, The Netherlands) and a pulse oximeter (Pulse Oximeter 8600, Nonin Medical Inc., Minneapolis, MN, USA). The peripheral oxyhemoglobin saturation (SpO_2_, %) was continuously measured and the difference (∆SpO_2_) between SpO_2_ peak, as mean SpO_2_ of the last 20 sec of the peak, and SpO_2_ rest, as the mean of the 3 min rest period, was recorded. Every 2 min, blood pressure was measured.

The test was considered as maximal when one of the following criteria was achieved: predicted peak oxygen uptake, predicted maximal work rate, predicted maximal heart rate, evidence of ventilatory limitation or respiratory exchange ratio (RER) value greater than 1.15 [14]. Criteria for terminating the exercise test were chest pain suggestive of ischemia, ischemic electrocardiogram changes, complex ectopy, second- or third-degree heart block, fall in systolic pressure > 20 mmHg from the highest value during the test, critical hypertension, severe desaturation, sudden pallor, loss of coordination, mental confusion, dizziness or faintness, signs of respiratory failure [15]. To calculate the predicted values, the equations by Wasserman et al. [16] were used.

We recorded the mean value of the peak workload (in watts) and of the peak V’O_2_ (in mL/kg/min) during the last 20 s of the test. V-slope and ventilatory equivalents methods (‘dual method approach’) [16] were both used to establish the anaerobic threshold (AT), which was expressed as the absolute value of V’O_2_ in mL/kg/min.

The formula VEmax/Maximum voluntary ventilation × 100 was used to calculate the breathing reserve (BR, %). Multiplying FEV_1_ by 40, we calculated the maximum voluntary ventilation [17]. The ventilatory response during exercise was expressed as a linear regression function by plotting VE against VCO_2_ obtained every 10 s, excluding data above the ventilatory compensation point [18]. Then, using the VE/VCO_2_ regression line, we obtained the slope and Y intercept values. The end-tidal pressure of CO_2_ (PETCO_2_, in mmHg) was measured as the mean of PETCO_2_ during the 3 min rest period and during the last 20 s of the test and was recorded as the difference between the PETCO_2_ peak and PETCO_2_ rest (∆PETCO_2_).

Changes in operational lung volumes were assessed every two minutes during exercise and at peak exercise, taking the IC measured at rest as the baseline. As described by Stubbing et al. [18], COPD patients maintain a constant TLC during exercise; therefore, changes in end-expiratory lung volume are related to changes in IC. Accordingly, we recorded IC at rest, at peak of exercise and the difference between IC peak and IC rest (∆IC).

The oxygen pulse (O_2_Pulse) at peak, the double product reserve (DPR) and the heart rate recovery at peak of exercise (HRR) reflect the cardiovascular response to exercise. O_2_Pulse (mL/bpm) was calculated by dividing instantaneous oxygen uptake by the heart rate [16]. As described by Le VV et al. [19], we calculated DPR (mmHg·bpm) by subtracting the double product, i.e., the product of systolic blood pressure and heart rate, at maximal exercise minus that at rest. HRR (bpm) was defined as the reduction in the HR from the peak exercise level to the rate 1 min after the end of exercise [20].

A visual analogue scale (VAS), scored from 0 to 100, was employed to ascertain the degree of dyspnea and leg fatigue induced by the incremental exercise. This scale consisted of a horizontal line with the words ‘none’ and ‘very severe’ placed at left and right, respectively. The assessments of perceived dyspnea and leg fatigue were subsequently divided by the maximum workload (VAS_dys_/WL and VAS_fat_/WL, in mm/watts) for analysis [21].

### 2.3. Statistical Analysis

Values are presented as mean ± standard deviation (SD) or as median (interquartile range). Given the explorative nature of the study, no formal sample size calculation was performed. The distribution of variables was assessed by the Kolmogorov–Smirnov goodness-of-fit test. Comparisons between variables were obtained by unpaired *t* test, ANOVA with post hoc test and Chi-square test, when appropriate. Correlations between variables were assessed by Pearson or Spearman correlation coefficient, when appropriate. Quade’s nonparametric ANCOVA test was used to analyze differences in VO_2_ peak (dependent variable) between active and inactive patients, adjusting for age, sex, BMI, mMRC and FEV_1_/FVC as covariates. *p* < 0.05 was taken as significant.

## 3. Results

In total, 214 consecutive patients with COPD (62 females) were recruited, with an age range between 41 and 85 years. According to the FEV_1_-predicted value, 14.9%, 44.1%, 34.8% and 6.2% of patients were classified as mild, moderate, severe and very severe patients, respectively. A total of 136 (63.5%) patients were ex-smokers, 67 (31.5%) current smokers and 11 (5.0%) never smokers. At the time of study entry, patients were receiving either regular therapy with inhaled steroids (58.0%), long-acting beta_2_-agonists (79.6%) and long-acting muscarinic antagonists (67.1%) or no treatment (17.4%).

A total of 83 (38.8%, females 25.3%, age range 41–85 y) and 131 (61.2%, females 31.3%, age range 49–83 y) out of 214 patients were categorized as active and inactive ones, respectively (Table 1). All the included patients performed spirometry and completed the CPET without any complications.

Active and inactive patients did not differ in sex distribution, age, BMI and in baseline lung function values (Table 1). They also did not differ in either smoking history, percentage of hypertensive patients or current treatment. According to the predicted FEV_1_ value, mild stage was the most prevalent in both active (54.3%) and inactive (41.0%) patients. Active and inactive patients were significantly different in mMRC values (Table 1); moreover, the percentage of inactive patients with an mMRC value ≥ 2 was significantly higher than the corresponding one of active patients (51.4% vs. 31.2%, Chi-square = 8.283, *p* = 0.004).

Moreover, the two groups were significantly different in peak V’O_2_ both as the absolute value and as a percentage of the predicted value as well as in maximal workload and in V’O_2_ at the AT. Active and inactive patients significantly differed in cardiovascular response to exercise (O_2_Pulse at peak, DPR and HRR), but not in ventilatory response and in changes in operational volumes during exercise, as assessed by the CPET (Table 2) (Figure 1). Significant differences in peak V’O_2_ both as the absolute value (F = 5.483, *p* = 0.020) and as a percentage of the predicted value (F = 6.125, *p* = 0.014) were found between active and inactive patients by means of Quade’s nonparametric ANCOVA test after adjusting for age, sex, BMI, mMRC and FEV_1_/FVC as covariates.

Active patients showed VAS_dys_/WL (dyspnea rating for workload) and VAS_fat_/WL (muscle fatigue rating for workload) values significantly lower, when compared to inactive ones (Table 2).

Lastly, in all patients, the V’O_2_ values in mL/kg/min both at peak and at the AT were significantly and positively related to the corresponding ones of the DPR (r = 0.517, *p* < 0.001 and r = 0.420, *p* < 0.001) and HRR (r = 0.282, *p* = 0.001 and r = 0.208, *p* = 0.032) (Figure 2). Additionally, both the VAS_dys_/WL and VAS_fat_/WL values were significantly and negatively related to the corresponding ones of the DPR (r = −0.430, *p* < 0.001 and r = −0.393, *p* < 0.001) and HRR (r = −0.340, *p* = 0.001 and r = −0.207, *p* = 0.003).

## 4. Discussion

Physical activity is defined as “any bodily movement produced by skeletal muscles, that results in energy expenditure” and is positively correlated with physical fitness [22]. Physical activity should not be confused with physical exercise, which is “a subset of physical activity that is planned, structured, and repetitive and has as a final or an intermediate objective the improvement or maintenance of physical fitness” [22]. By assessing the maximal capability of patients to perform exercise, the reserve capacity of each of the organ systems contributing to the exercise response, such as the circulation and respiratory systems, may be evaluated [5]. Previous studies showed moderate [23] or no [24] relationship between physical activity and exercise capacity in patients with COPD, without reporting any information on the ventilatory and cardiovascular response to exercise of these patients. In the present study, we assessed the relationship between regular daily physical activity and exercise capacity in a large cohort of COPD patients, by addressing the reserve of the cardiopulmonary system.

A first expected result of this study is that most of the COPD patients engage in daily physical activity below the recommended threshold for maintaining or developing fitness. In addition, we found that active COPD patients had better cardiometabolic parameters when undergoing maximal exercise, as compared to inactive patients. Notably, active patients showed oxygen uptake values both at the peak of exercise and at the anaerobic threshold higher than inactive patients, in spite of the active and inactive patients not differing either in anthropometric characteristics or in resting function and in ventilatory response and in changes in operational volumes during exercise. It is of note that the active patients experienced less daily living activity-related dyspnea and showed better values of O_2_Pulse at peak, DPR and HRR than inactive patients. In addition, in all COPD patients, the oxygen uptake values both at the peak of exercise and at the anaerobic threshold were positively related to cardiovascular response in terms of both the DPR and HRR. Lastly, we also found that dyspnea and leg fatigue perceptions induced by incremental exercise were negatively related to DPR and HRR values.

The O_2_Pulse can be considered as an indirect marker of stroke volume, when arterial oxygen content is assumed normal [25]. The DPR is an estimate of the maximal performance of the left ventricle. Notably, the DPR reflects myocardial oxygen uptake during exercise; this is because the three primary determinants of myocardial oxygen uptake include ventricular wall tension, heart contractility and heart rate [18]. In COPD patients with different degrees of severity, dynamic hyperinflation is strongly linked to an inadequate cardiovascular response to exercise in terms of both the O2Pulse and DPR [26]. HRR was identified as a powerful predictor of mortality in the general population, independent of the workload and the HR changes during exercise [20]. A decrease in HR of 12 beats per minute or less at 1 min after peak exercise was associated with a higher risk of mortality from any cause over a 6-year period [20]. Moreover, a low HRR was found to be associated with a reduced survival in COPD patients [27].

There is a body of evidence on the beneficial effects of regular physical activity in increasing cardiovascular performance. By contrast, a sedentary lifestyle significantly elevates the risk of cardiac morbidity and mortality. Findings from the Honolulu Heart Program, which targeted physically capable elderly men, showed that the risk of coronary heart disease is reduced with increases in the daily walked distance [28]. Similar results were found in a large cohort of healthy women aged 45 or older [29]. It is known that regular aerobic activity may lower the heart rate and resting blood pressure; this allows the heart to reduce the workload and be less exposed to risks of overload diseases [1].

The findings of the present study are to be interpreted in the context of the limitations. The first limitation is due to the retrospective nature of the study, which did not allow having sufficient data on the assessment of quality of life using specific questionnaires, nor a control group. In addition, no causality or directionality of the findings can be inferred, since the study is a cross-sectional study. Another limitation is because we assessed the DPA in our patients by means of a questionnaire, patients could overestimate or underestimate their DPA. A strength of the study is the recruitment of a wide cohort of patients, ranging from mild to moderately severe disease, which reflects the general population of patients with COPD.

## 5. Conclusions

In summary, this study shows that active patients, while not differing from inactive patients in anthropometric and pulmonary function characteristics, may have a higher exercise capacity due to better cardiovascular function. This study further suggests the beneficial effects of regular physical activity in COPD. Health professionals and policy makers should implement programs, practices and policies to facilitate increased physical activity in COPD patients as well as in the general population.

## Figures and Tables

**Figure 1 medicina-60-01026-f001:**
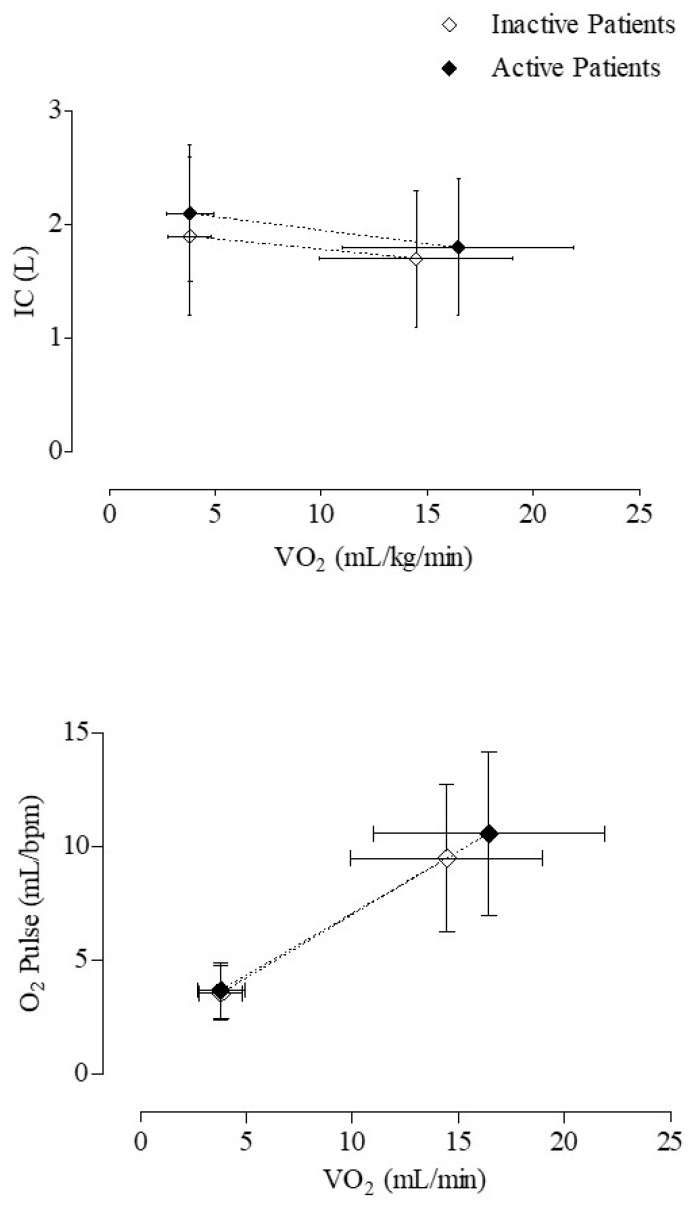
Mean and standard deviation values at rest and peak of the inspiratory capacity (**upper panel**) and O_2_Pulse (**lower panel**) in relation to the corresponding rest and peak VO_2_ values in 83 active and in 131 inactive patients with COPD. Mean IC values at rest and at peak of exercise did not differ between active and inactive patients, whereas mean O_2_Pulse and VO_2_ values were significantly higher at peak of exercise (*p* < 0.01), but not at rest in active patients as compared to inactive ones.

**Figure 2 medicina-60-01026-f002:**
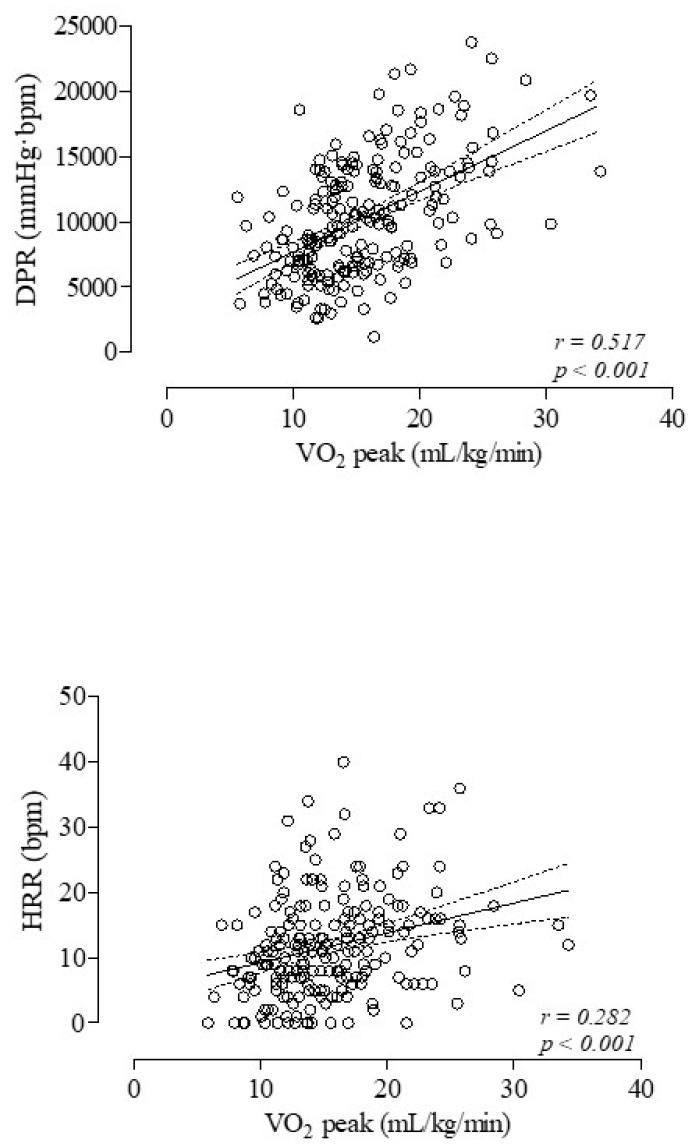
Relationships between V’O_2_ values in mL/kg/min at peak of exercise and double product reserve (DPR) (**upper panel**) and heart rate recovery at peak of exercise (HRR) (**lower panel**) in 214 COPD patients.

**Table 1 medicina-60-01026-t001:** Anthropometric, clinical and resting lung function characteristics of 214 chronic obstructive pulmonary disease (COPD) patients.

Parameter	All Patients N. 214 (100%)	Active Patients N. 83 (38.8%)	Inactive Patients N. 131 (61.2%)
Age (yr)	68.2 ± 8.6	68.0 ± 8.7	68.2 ± 8.0
M/F (N.) (F%)	152/62 (28.9)	62/21 (25.3)	90/41 (31.3)
BMI (Kg/m^2^)	27.4 ± 5.2	27.2 ± 5.1	27.5 ± 5.3
mMRC (0–4)	1 (1–2)	1 (1–2)	2 (1–2) **
TLC (L)	6.810 ± 1.346	6.808 ± 1.362	6.812 ± 1.339
TLC (% predicted)	115.2 ± 19.6	114.0 ± 19.4	116.1 ± 19.7
FVC (L)	2.842 ± 0.756	2.883 ± 0.704	2.816 ± 0.790
FVC (% predicted)	85.7 ± 17.6	86.3 ± 17.2	85.3 ± 17.9
FEV_1_ (L)	1.428 ± 0.592	1.504 ± 0.589	1.380 ± 0.590
FEV_1_ (% predicted)	56.2 ± 19.3	58.4 ± 18.4	54.7 ± 19.7
FEV_1_/FVC (%)	47.3 ± 12.5	48.3 ± 11.3	46.6 ± 13.1
IC (L)	2.02 ± 0.64	2.13 ± 0.58	1.95 ± 0.67
IC/TLC (%)	31.7 ± 9.5	33.1 ± 9.6	30.1 ± 9.4
DLCO (% predicted)	63.5 ± 23.3	65.5 ± 23.0	62.2 ± 23.5

Values are expressed as mean ± SD, as median (25–75% percentile) or as percentage; ** *p* < 0.01 active vs. inactive patients. Abbreviations. M: males, F: females, BMI: body mass index, mMRC: modified Medical Research Council, TLC: total lung capacity, FVC: forced vital capacity, FEV_1_: forced expiratory volume at 1st second, IC: inspiratory capacity, DLCO: diffusion lung capacity for CO.

**Table 2 medicina-60-01026-t002:** CPET parameters of 214 COPD patients.

Parameter	All Patients N. 214 (100%)	Active Patients N. 83 (39%)	Inactive Patients N. 131 (61%)
Workload (watts)	75.8 ± 35.5	86.8 ± 38.9	68.9 ± 31.4 **
V’O_2_ peak (mL/kg/min)	15.4 ± 4.9	16.9 ± 5.6	14.5 ± 4.3 **
V’O_2_ peak (% pred)	64.6 ± 21.5	70.9 ± 25	60.6 ± 17.7 **
V’O_2_ @AT (mL/kg/min)	11.5 ± 3.9	12.5 ± 4.5	10.9 ± 3.4 *
V’O_2_/HR peak (mL/bpm)	10.0 ± 3.4	10.8 ± 3.6	9.5 ± 3.1 **
V’O_2_/HR peak (% pred)	83.7 ± 26.0	89.9 ± 29.5	79.8 ± 22.8 **
DPR (mmHg·bpm)	10,196 ± 4454	11,203 ± 4986	9557 ± 3970 **
HRR (bpm)	12.6 ± 12.2	13.0 ± 5.1	9.9 ± 7.0 *
BR (%)	74.0 ± 18.0	74.1 ± 17.7	75.1 ± 21.8
VE/V’CO_2_ slope	33.2 ± 8.8	33.4 ± 10.1	34.3 ± 8.1
VE/V’CO_2_ intercept	3.7 ± 2.6	3.3 ± 2.5	3.9 ± 2.7
∆PETCO_2_ (mmHg)	5.4 ± 4	5.7 ± 5	5.2 ± 4
∆SpO_2_ (%)	−1.8 ± 2.8	−1.7 ± 2.9	−1.9 ± 2.7
∆IC (L)	−0.251 ± 0.424	−0.260 ± 0.437	−0.245 ± 0.416
VAS_dys_/WL (mm/watts)	1.22 ± 0.7	1.05 ± 0.6	1.32 ± 0.8 **
VAS_fat_/WL (mm/watts)	1.13 ± 0.7	0.96 ± 0.6	1.23 ± 0.7 **

Values are expressed as mean ± SD; * *p* < 0.05, ** *p* < 0.01 active vs. inactive patients. Abbreviations. VO_2_: O_2_ uptake at peak, AT: anaerobic threshold, HR: heart rate, DPR: double product reserve, HRR: heart rate recovery, BR: breathing reserve, VE: minute ventilation, V’CO_2_: carbon dioxide production, PETCO_2_: end-tidal pressure of CO_2_, SpO_2_: peripheral oxyhemoglobin saturation, IC: inspiratory capacity, VAS: visual analogue scale, VAS_dys_/WL (dyspnea rating for workload) and VAS_fat_/WL (muscle fatigue rating for workload).

## Data Availability

The data that support the findings of this study are available on request from the corresponding author, A.M. The data are not publicly available due to information that could compromise the privacy of research participants.

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
