# Peer review of "Effects of Daily Physical Activity on Exercise Capacity in Chronic Obstructive Pulmonary Disease"

_medicina, 2024, doi:10.3390/medicina60071026_

Round 1

Reviewer 1 Report

Comments and Suggestions for Authors

Effects of daily physical activity on exercise capacity in chronic obstructive pulmonary disease

 The authors presented a well written and oragnized work. However, I suggest the below minor points:

 Beginning of abstract is the same as beginning of introduction. Perhaps paraphrase.

 Hypothesis of the study? I suggest reporting only the aim of the study without mentioning hypothesis.

 Methods:

Why did you use Rapid Assessment of Physical Activity, but not IPAQ?

Who supervised tests Spirometry and CPET? Was a physician present during the CPET? Please add

What was the COPD stage of the included participants? Did you take only a stable clinical condition (without exacerbation for at least one month) participants without knowing in which stage they are?

 Results:

The quality of figures is low

Reviewer 2 Report

Comments and Suggestions for Authors

Thank you for the opportunity to review the manuscript titled “Effects of daily physical activity on exercise capacity in chronic obstructive pulmonary disease”.

The authors explored the relationship between the amount of daily physical activity and exercise capacity in patients with COPD by retrospectively gauging the gross daily physical activity based on questionnaire responses and CPET performance. However, there are several deficiencies in the study that limit its validity.

1.      The authors did not specify what kind of COPD patients were included (although a short statement of “ranging from mild to moderately severe disease” was mentioned in the last paragraph of the discussion) and I assumed that they have included a full spectrum of COPD patients and the study was performed across a relatively long time span of 10 years. The authors should select either moderate to very severe COPD or separately analyze the exercise performance according to different COPD severities. The treatment of COPD has been substantially changed in the past 10 years. Patients may receive treatment differently, which would affect their exercise performance. The exercise tolerance or CPET performance would be different for COPD patients receiving inhaled corticosteroids only and those who received dual bronchodilators, even with similar FEV1 (or COPD groupings)

2.      Although the authors have mentioned that comorbidities affecting exercise performance would be excluded, they did not provide additional information about the comorbidities profile. Patients with chronic liver or renal impairment could have physical inactivity due to various physical discomforts. Additional cardiac assessment or lung function parameters at the baseline including PASP, LVEF and DLCO should be provided

3.      The authors should perform statistical analysis to adjust the effect of age, FEV1, time between last exacerbation and CPET assessment and other variables for the differences in exercise performance

4.      The authors should provide information on whether they have collected additional respiratory assessment including SGRQ or TDI

5.      The study value would be raised if there is further comparison with a group of non-COPD participants with CPET results and DPA data

6.      As this is a cross-sectional study, the last sentence of the conclusion in the abstract and similar statements in the main text should be removed or carefully revised

Comments on the Quality of English Language

The quality of English should be improved. 

Round 2

Reviewer 2 Report

Comments and Suggestions for Authors

Thank you for the opportunity to review the revised manuscript titled “Effects of daily physical activity on exercise capacity in chronic obstructive pulmonary disease”.

 The authors revised the manuscript based on the reviewers’ comments. However, I would still need the following issues be clarified or revised.

 1.      The authors need to carefully specify the patients recruited into the study. Were all patients or only randomly selected patients in the study period (2014 to 2023) fulfilling the inclusion / exclusion criteria included in the study?

2.      Line 82 to 85. The authors should arrange the sentences so that physical activities of different intensities are presented in ascending or descending order.

3.      Units should be presented in the full form in the main text (e.g. “s” in line 142, “min” in 156)

4.      Line 159: “TLC” instead of “total lung capacity”

5.      Line 197 to 199. The sentence structure should be revised, i.e., 63.5%, 31.5%, and 5.0% were ex-smokers, current smokers and never smokers (exact numbers of patients should be presented before the percentages).

6.      The decimal places should be standardized throughout the whole manuscript. For example, the percentages with one decimal place should be presented in the first paragraph of the results session. Similar issues in the Tables 1 (except mMRC grading) and 2

7.      The authors need to explain why the drug history for hypertension is considered essential in the first paragraph of the results session

8.      Table 1. Gender should be presented with a numerical number and percentage for either male or female, but not both genders

9.      Table 1. “predicted” instead of “pred”

10.  Line 256, remove “shown in”

Comments on the Quality of English Language

minor editing is required
